# Synthesis, Thermal Properties and Decomposition Mechanism of Poly(Ethylene Vanillate) Polyester

**DOI:** 10.3390/polym11101672

**Published:** 2019-10-14

**Authors:** Alexandra Zamboulis, Lazaros Papadopoulos, Zoi Terzopoulou, Dimitrios N. Bikiaris, Dimitra Patsiaoura, Konstantinos Chrissafis, Massimo Gazzano, Nadia Lotti, George Z. Papageorgiou

**Affiliations:** 1Laboratory of Chemistry and Technology of Polymers and Dyes, Department of Chemistry, Aristotle University of Thessaloniki, GR-54124 Thessaloniki, Macedonia, Greece; azampouli@chem.auth.gr (A.Z.); lazaros.geo.papadopoulos@gmail.com (L.P.); terzoe@gmail.com (Z.T.); 2Solid State Physics Department, School of Physics, Aristotle University of Thessaloniki, GR-54124 Thessaloniki, Greece; dpatsi@physics.auth.gr (D.P.); hrisafis@physics.auth.gr (K.C.); 3Organic Synthesis and Photoreactivity Institute, ISOF-CNR, Via Gobetti 101, 40129 Bologna, Italy; massimo.gazzano@unibo.it; 4Civil, Chemical, Environmental and Materials Engineering Department, University of Bologna, via Terracini 28, 40131 Bologna, Italy; nadia.lotti@unibo.it; 5Chemistry Department, University of Ioannina, P.O. Box 1186, 45110 Ioannina, Greece

**Keywords:** poly(ethylene vanillate), synthesis, thermal properties, crystallization, thermal stability, decomposition mechanism

## Abstract

Plastics are perceived as modern and versatile materials, but their use is linked to numerous environmental issues as their production is based on finite raw materials (petroleum or natural gas). Additionally, their low biodegradability results in the accumulation of microplastics. As a result, there is extensive interest in the production of new, environmentally friendly, bio-based and biodegradable polymers. In this context, poly(ethylene vanillate) (PEV) has a great potential as a potentially bio-based alternative to poly(ethylene terephthalate); however, it has not yet been extensively studied. In the present work, the preparation of PEV is reported. The enthalpy and the entropy of fusion of the pure crystalline PEV have been estimated for the first time. Additionally, the equilibrium melting temperature has also been calculated. Furthermore, the isothermal and non-isothermal crystallization behavior are reported in detail, and new insights on the thermal stability and degradation mechanism of PEV are given.

## 1. Introduction

Plastics are perceived as modern and versatile materials in the 21st century, and imagining our life without them seems impossible. Those materials provide many advantages for today’s society, fostering and enabling its sustainability [1]. However, plastics are also polluting materials. Indeed, their disposable character and short-term applications in combination with a low or absent biodegradability have led to the accumulation of microplastics worldwide, causing important environmental issues [2]. Additionally, their production depends on petroleum and natural gas, which are finite raw materials. As a result, environmental concerns have been raised both for the resources used and their end-life options [3]. In a quest towards greener materials, many natural feedstocks have been explored for the production of monomers that can be used to synthesize bio-based polymers, mainly cellulose, lignin and polysaccharides [4]. Therefore, environmentally friendly materials, bio-based and biodegradable, are intensively sought by the academic, as well as the industrial community to replace traditional polymers.

The replacement of fossil fuels for the production of monomers by alternative inexpensive and renewable starting materials, such as cellulose, starch, lignin, proteins and vegetable oils is a great challenge. The idea of producing polymers from renewable resources is not new. However, there is always a drawback, due to their relatively high cost in comparison with their petrochemical homologues. The cost of bio-based chemicals as building blocks is expected to decrease soon, and the production of platform chemicals and monomers following the concept of the biorefinery seems to be an answer. Biomass-derived monomers are generally classified on the basis of their natural molecular biomass origins into: (i) Oxygen-rich monomers, including carboxylic acids, polyols, dianhydroalditols, and furans; (ii) hydrocarbon-rich monomers, such as vegetable oils, fatty acids, terpenes, terpenoids and resin acids; (iii) hydrocarbon monomers, such as bio-ethene, bio-propene, bio-isoprene and bio-butene; and (iv) non-hydrocarbon monomers, i.e., carbon dioxide and carbon monoxide.

Replacing the widely used, petroleum-derived plastic poly(ethylene terephthalate) (PET) with a bio-based counterpart is an intriguing case study. PET is a widely used polymer, especially for short-term packaging applications. Along with the recycling of PET [5], the search for its bio-based alternatives has attracted considerable interest. Poly(ethylene furanoate) (PEF), and 2,5-furandicarboxylate polyesters are, in general, some of the most promising alternatives which have been extensively studied [6]. The reasons behind that interest are the facts that PEF exhibits superior thermal stability, a lower melting temperature, significantly lower O_2_ and CO_2_ permeability than PET, as well as excellent mechanical properties and good processability. Therefore, a series of studies have been performed concerning furan polyesters, focusing on key parameters, such as catalysis [7], solid state polymerization [8,9,10], copolymerization [11,12] and nanocomposite materials [13,14]. However, a very important factor that has to be taken into consideration is the high cost linked to the production of 2,5-furandicarboxylic acid [15]. It is a major drawback that hampers the further development of this class of polyesters, and therefore, other routes towards bio-based PET alternatives need to be also explored.

An interesting alternative bio-based building block which has recently regained attention is vanillic acid or 4-hydroxy-3-methoxybenzoic acid. Vanillic acid can be produced by oxidizing vanillin, which can be isolated from lignin [16,17]. Lignin is found in plant biomass; it is the second most abundant organic polymer in nature (after cellulose) and is a unique source of aromatic building blocks. It is not a coincidence that according to the BIOSPRI tender "Study on support to R&I policy in the area of bio-based products (BBPs) and services", implemented by the University of Bologna and the Fraunhofer ISI, seven of the top 20 selected bio-based products are derived from lignin [18]. Vanillin production is dominated by the catechol–guaiacol process, which is petroleum-dependent. However, Borregaard, the second largest vanillin producer worldwide, produces vanillin from lignin by oxidation (15% of the global vanillin production). In parallel, progress in lignin depolymerization [19] and the biotechnological production of vanillin [20,21], are expected to bring about more sustainable processes for vanillin production, rendering vanillin a top-priority renewable building block [22].

Poly(ethylene vanillate) (PEV) can be prepared from vanillic acid and 2-chloroethanol. As 2-chloroethanol can be produced from ethylene glycol, which can originate from biomass, PEV is considered a bio-based polymer. Due to its analogous aliphatic/aromatic structure, and also similar mechanical and thermal properties, PEV is considered a potential alternative to PET. However, despite its great potential, it has not yet been widely investigated. Mialon et al. investigated a series of polyesters derived from 4-hydroxybenzoic acid, vanillic acid and syringic acid (4-hydroxy-3,4-dimethoxybenzoic acid) [23]. The aromatic acids were modified with ω-chloro-alcohols on the aromatic hydroxyl group to afford hydroxy-carboxylic acids, which were further polymerized in the presence of Sb_2_O_3_ as a catalyst. Poly(alkylene vanillate)s exhibited higher melting temperatures (*T_m_*) than the corresponding poly(alkylene 4-hydroxybenzoate)s and poly(alkylene syringate)s and a decrease in the glass transition temperature (*T_g_*) was observed with the increasing length of alkylene chain. The polymerization of vanillic acid was unsuccessful, yielding a low-molecular weight, insoluble material. More recently, Gioia et al. reported a one-pot synthesis of PEV from vanillic acid in the presence of ethylene carbonate, catalyzed by dibutyltin oxide [24]. The average molecular weight of the polymers obtained in the latter work was slightly lower than in the work of Mialon et al. (4700 vs 5390 g mol^−1^), but the melting temperature rather higher (264 °C *vs* 239 °C). Gioia et al. copolymerized vanillic acid with ε-caprolactone [24] and ricinoleic acid [25], while Nguyen et al. have reported the copolymerization of 4-hydroxyethylvanillic acid with ε-caprolactone and l-lactide [26]. Apart from PEV, other vanillic polymers have been synthesized, as the aromatic units tend to increase the mechanical properties and the *T_g_* of the resulting polymers [27,28,29,30,31]. Finally, vanillic acid has also been incorporated in thermotropic liquid crystalline polyesters [32,33,34,35,36].

PEV is a new bio-based polyester for promising applications mainly as a packaging material. However, extensive studies on its thermal properties have not yet been reported. In the present study, PEV was synthesized via the melt polycondensation of 4-(2-hydroxyethoxy)-3-methoxybenzoic acid or 4-hydroxyethylvanillic acid. The thermodynamic properties, such as the enthalpy and entropy of fusion of 100% crystalline PEV and the equilibrium melting temperature were estimated and are reported for the first time. The multiple melting behavior was also investigated. The isothermal crystallization from the melt, as well as the non-isothermal crystallization from the glass and from the melt, were studied using differential scanning calorimetry (DSC), polarized light microscopy (PLM), and wide-angle X-ray diffraction (WAXD). Furthermore, thermogravimetric analysis (TGA) and pyrolysis-gas chromatography/mass spectroscopy (Py-GC/MS) measurements were also conducted, giving new insights about the thermal stability of PEV and its decomposition mechanism.

## 2. Materials and Methods

### 2.1. Materials

Vanillic acid (VA, purum 97%), 2-chloroethanol (>99%), titanium butoxide (Ti(OBu)_4_) antimony trioxide (Sb_2_O_3_ 99.99%) catalyst were purchased from Aldrich Co. (Chemie GmbH, Steinheim, Germany). The purchased monomers have a petrochemical origin.

### 2.2. Synthesis of 4-(2-Hydroxyethoxy)-3-Methoxybenzoic Acid

16.0 g (0.095 mol) of vanillic acid and 2.85 g (0.019 mol) of sodium iodide were dissolved in aqueous sodium hydroxide (15.2 g (3.81 mol) in 70 mL of water); 11.49 g (0.14 mol) of chloroethanol dissolved in 140 mL of ethanol and degassed by N_2_ bubbling were added dropwise at 100 °C. The reacting mixture was refluxed, and 3.8 g of chloroethanol were added every 24 h. After 4 days, the reacting mixture was concentrated, and the residue was dissolved in water. The aqueous solution was washed with ether and acidified with HCl(aq.). The solid that precipitated was isolated by filtration and purified by recrystallisation in ethanol to afford the product in 54% yield. ^1^H NMR (500 MHz, DMSO-*d*_6_, δ): 12.64 (s, 1H), 7.54 (dd, *J* = 8.5, 2.0 Hz, 1H), 7.45 (d, *J* = 2.0 Hz, 1H), 7.04 (d, *J* = 8.5 Hz, 1H), 4.89 (s, 1H), 4.04 (t, *J* = 4.9 Hz, 2H), 3.80 (s, 3H), 3.74 (t, *J* = 4.9 Hz, 2H). ^13^C NMR (126 MHz, DMSO-*d*_6_, δ): 167.1, 152.1, 148.4, 123.2, 122.9, 112.1, 111.9, 70.2, 59.4, 55.4.

### 2.3. Synthesis of Poly(Ethylene Vanillate) (PEV)

PEV was prepared by a two-stage melt polycondensation procedure in a glass batch reactor [6,37]. According to this, in the first esterification step, 8 g of 4-(2-hydroxyethoxy)-3-methoxybenzoic acid, which was produced and purified previously, were added into the reaction tube of the polycondensation reactor. Ti(OBu)_4_ (TBT) 400 ppm or Sb_2_O_3_ was charged, and the apparatus containing the reagents was evacuated several times and filled with nitrogen in order to remove the existing oxygen. The reagents were heated at 190 °C under argon flow (50 mL/min) for 2 h.

In the second step of polycondensation, vacuum (5.0 Pa) was applied slowly over a period of time of about 15 min, and the temperature was gradually increased to 240 °C, while stirring speed was also increased from 350 rpm to 720 rpm. The reaction continued at this temperature for 2 h. Then, the temperature was gradually increased to 250 °C and 260 °C for 1h each step. After the polycondensation reaction was completed, the polyesters were removed from the reactor, milled and washed with methanol. The polycondensation was followed by solid state polymerization (SSP), that was performed under vacuum in a glass batch reactor. The milled polymer obtained after the melt polycondensation was heated under vacuum for 5 h, at 230 °C for 30 min, at 240 °C for 1h, at 250 °C for 1.5 h and at 255 °C for 2 h.

### 2.4. Polyester Characterization

#### 2.4.1. Intrinsic Viscosity Measurement

Intrinsic viscosity [*η*] measurements were performed with an Ubbelohde viscometer (Schott Gerate GMBH, Hofheim, Germany) at 25 °C in a mixture of phenol and tetrachloroethane (60/40, *w*/*w*). The samples were maintained in the above mixture of solvents at 60 °C for 20 min to achieve complete dissolution. The intrinsic viscosity of polyester was calculated using the Solomon–Ciuta Equation (1) of a single point measurement:(1)[η] = [2{tt0−ln(tt0)−1}]12c,
where *c* is the concentration of the solution; *t*, the flow time of solution and *t*_0_ the flow time of pure solvent. For each sample, three measurements were conducted, and the average value was calculated.

#### 2.4.2. Wide Angle X-ray Diffraction Patterns (WAXD)

X-ray diffraction measurements of the samples were performed using a MiniFlex II XRD system from Rigaku Co. (Rigaku Co., Tokyo, Japan), with CuK_α_ radiation (λ = 0.154 nm) in the angle (2*θ*) range from 5 to 65 degrees.

#### 2.4.3. Fourier Transformed-Infrared Spectroscopy (FTIR)

FTIR spectra were obtained using a Perkin–Elmer FTIR spectrometer (Perkin Elmer, Waltham, MA, USA), model Spectrum One, in absorbance mode and in the spectral region of 500–4000 cm^−1^ using a resolution of 4 cm^−1^ and 64 co-added scans.

#### 2.4.4. Nuclear Magnetic Resonance (NMR)

NMR spectra were recorded in deuterated dimethylsulfoxide (DMSO-d_6_) or trifluoroacetic acid (TFA-d), on an Agilent 500 spectrometer (Agilent Technologies, Santa Clara, CA, USA), at room temperature. Spectra in TFA-d were recorded in the presence of a DMSO probe Spectra were internally referenced with tetramethylsilane (TMS) and calibrated using the residual solvent peaks.

#### 2.4.5. Differential Scanning Calorimetry (DSC)

Thermal analysis studies were carried out using a using Perkin Elmer Diamond DSC (PerkinElmer Corporation, Waltham, MA, USA) updated to DSC 8500 level, combined with an Intracooler IIP cooling system. Samples of about 5 mg were used. The samples were first heated at 20 °C/min to a temperature 40 °C above the melting peak temperature, to erase the previous thermal history. To record the glass transition, the samples were first melt-quenched on a frozen metal plate and then were put in the DSC cell. In general, heating scans of the samples were conducted at 20 °C/min. To study the crystallization on cooling from the melt, scans at different rates were performed.

Isothermal crystallization experiments of the polymers at various temperatures below the melting point were performed after self-nucleation (SN) of the polyester sample. The SN procedure leads to enhanced crystallization rates, so that it allows crystallizations to be performed even in case of very small supercoolings, that is at temperatures close to the melting temperature. As a result, crystallizations were tested in a wide temperature range. Self-nucleation measurements were performed in analogy to the procedure described by Fillon et al. [38]. The protocol used was very similar with that described by Müller et al. [39], and can be summarized as follows: (a) Melting of the sample at 40 °C above the observed melting point for 5 min to erase any previous thermal history; (b) cooling at 10 °C/min to a reference temperature and crystallization, to create a ‘‘standard’’ thermal history; (c) partial melting by heating at 20 °C/min up to a “self-nucleation temperature”, *T_s_* which differed for the various polymers; and (d) thermal conditioning at *T_s_* for 5 min. Depending on the *T_s_*, the crystalline polyester will be completely molten, only self-nucleated or self-nucleated and annealed. If *T_s_* is sufficiently high, no self-nuclei or crystal fragments can remain. At intermediate *T_s_* values, the sample is almost completely molten, but some small crystal fragments or crystal memory effects remain, which can act as self-nuclei during a subsequent cooling from *T_s_*. Finally, if *T_s_* is too low, the crystals will only be partially molten, and the remaining crystals will undergo annealing during the 5 min at *T_s_*, while the molten crystals will be self-nucleated during the following cooling; (e) cooling scan from *T_s_* at 20 °C/min to the crystallization temperature (*T_c_*), where the effects of the previous thermal treatment will be reflected on isothermal crystallization; (f) heating scan at 20 °C/min to 40 °C above the melting point, where the effects of the thermal history will be apparent on the melting signal. Experiments were performed to check that the sample did not crystallize during the cooling to *T_c_* and that a full crystallization exothermic peak was recorded at *T_c_*. In heating scans after isothermal crystallization, the standard heating rate was 20 °C/min. In some specific cases, heating experiments at different rates after isothermal crystallization was also carried out to better understand the melting behavior of the polyester; and for PEV *T*_s_ was 265 °C. Tests showed that at higher *T_s_* values, the polyester was completely molten. In contrast, annealing was observed at temperatures lower than 265 °C.

To investigate the non-isothermal crystallization of the polyesters from the melt, the samples were first heated to 310 °C for 1 min and then the samples were cooled from the melt under a wide range of cooling rates, from 5 to 20 °C/min. To study the cold-crystallization behavior the samples were first melt-quenched and then heated at the predetermined heating rate (5, 10, 15, 20 °C/min).

#### 2.4.6. Polarizing Light Microscopy (PLM)

A polarizing light microscope (Nikon, Optiphot-2, Melville, NY, USA) equipped with a Linkam THMS 600 heating stage (Linkam Scientific Instruments Ltd., Surrey, UK), a Linkam TP 91 control unit and also a Jenoptic ProgRes C10Plus camera (Jenoptik Optical Systems GmbH, Jena, Germany) were used for PLM observations.

#### 2.4.7. Thermogravimetric Analysis (TGA)

Thermogravimetric analysis was performed with a SETARAM SETSYS TG-DTA 16/18 instrument (Setaram instrumentation, Lyon, France). The samples (8 ± 0.2 mg) were placed in alumina crucibles, while a blank measurement was performed and subsequently was subtracted by the experimental curve, in order to eliminate the buoyancy effect. PEV samples were heated from ambient temperature up to 550 °C in a 50 mL/min N_2_ flow at the following heating rates—5, 10, 15, and 20 °C/min. Continuous recording of both sample temperature and sample weight was carried out.

#### 2.4.8. Pyrolysis-Gas Chromatography-Mass Spectroscopy (Py-GC/MS)

For Py-GC/MS analysis of the polyesters, a very small amount of each material is “dropped” initially into the “Double-Shot” EGA/PY-3030D Pyrolyzer (Frontier Laboratories Ltd., Fukushima Japan) using a CGS-1050Ex (Japan), carrier gas selector. For pyrolysis analysis (flash pyrolysis), each sample was placed into the sample cup, which afterwards fell free into the pyrolyzer furnace. The pre-selected pyrolysis temperatures were 330 °C, 420 °C and 500 °C the GC oven temperature was heated from 50 to 300 °C at 20 °C/min. Those two temperatures were selected based on the TGA pyrogram and represent the sample prior and after thermal decomposition. Sample vapors generated in the furnace were split (at a ratio of 1/50), a portion moved to the column at a flow rate of 1 mL/min, pressure 53.6 kPa and the remaining portion exited the system via the vent. The pyrolyzates were separated using temperature-programmed capillary column of a Shimadzu QP-2010 Ultra Plus (Shimadzu, Kyoto, Japan) gas chromatogram and analyzed by the mass spectrometer MS-QP2010SE of Shimadzu (Japan) use 70 eV. Ultra-ALLOY^®^ metal capillary column from Frontier Laboratories LTD (Fukushima, Japan) was used containing 5% diphenyl and 95% dimethylpolysiloxane stationary phase, column length 30 m and column ID 0.25 mm. For the mass spectrometer the following conditions were used: Ion source heater 200 °C, interface temperature 300 °C, vacuum 10^-4^–10^0^ Pa, *m*/*z* range 10–500 amu and scan speed 10,000. The chromatogram and spectra retrieved by each experiment were subject to further interpretation through Shimadzu and Frontier post-run software.

#### 2.4.9. Nanoindentation

PEV was assessed through nanoindentation tests in order to measure its modulus and hardness. In instrumented indentation tests, the load is measured as a function of penetration depth. Such tests enable local variations of modulus and hardness to be measured precisely [40,41,42,43]. In the current work, the indentations were conducted using a dynamic ultra-micro-hardness tester (DUH-211; Shimadzu Co., Kyoto, Japan) fitted with a triangular pyramid indenter tip (Berkovich indenter). The indentations made on the surface of PEV film (about 1 mm depth prepared at 270 °C) appeared as an equilateral triangle. Ten measurements were conducted on each sample, which were purposely scattered on the surface. After contact of the indenter with the surface, this was driven into the surface until a peak load of 500 mN was reached. The peak load was held for 3 s (in order to minimize the effect of viscoelastic deformation of the specimen, notably creep, on property measurements) and then the indenter was unloaded, to a load of zero.

## 3. Results and Discussion

### 3.1. Synthesis and Structural Characterization of PEV

4-(2-Hydroxyethoxy)-3-methoxybenzoic acid was synthesized from vanillic acid and 2-chloro-1-ethanol via a Williamson reaction, according to the literature [23]. PEV was synthesized via the two-step polycondensation method, according to the reaction procedure presented in Scheme 1. The received material was solid, and the color was light yellow. The intrinsic viscosity values of the prepared polyesters were 0.28 and 0.32 g/dL using TBT and Sb_2_O_3_ as catalysts, respectively. After the SSP procedure, their intrinsic viscosity values were increased to 0.34 and 0.38 g/dL, respectively.

The synthesized PEV was characterized by FT-IR and NMR, and both methods confirmed the successful polymerization of 4-(2-hydroxyethoxy)-3-methoxybenzoic acid. In the IR spectra, a decrease in the OH band at 3400 cm^−1^ is clearly visible (Figure 1). Even more strikingly, the bands at 2533 and 2626 cm^−1^, attributed to the OH groups bound to COOH moieties through H-bonds [44], were diminished in the spectra of the polymers, indicating successful polymerization. Finally, the shift of the band corresponding to the C=O bond from 1681 cm^−1^ in the monomer to 1713 cm^−1^ in the polymers further confirmed those observations. In PEV, after SSP, the peak attributed to –OH groups is further reduced, due to their reaction with –COOH end groups and molecular weight increase.

Similarly to FT-IR, the NMR spectra showed that a successful polymerization was carried out (Figure 2). This was mainly inferred by the shift of the methylene protons of the O–CH_2_–CH_2_–OH segment from 4.09 and 4.20 ppm to 4.40 and 4.68 ppm, as a result of the esterification of the hydroxyl group. Corresponding changes are also observed in the ^13^C spectra, and, in addition, the peak corresponding to the COOH carbon at 171.8 ppm disappeared and was replaced by a peak at 168.1 ppm corresponding to the carbon of the ester moieties.

### 3.2. Thermal Properties and Crystallization Behavior of PEV

The synthesized PEV samples were studied with respect to their thermal transitions with DSC (Figure 3). The low molecular weight of the as-prepared polyester synthesized with TBT catalyst after solid state polymerization showed a melting peak temperature *T_m_ =* 261 °C (Figure 3a). After quenching, the amorphous sample showed a *T_g_ =* 75 °C and a sharp cold crystallization peak at *T_cc_ =* 107 °C. On cooling at 10 °C/min, a broad peak appeared at 145 °C. On the contrary, the PEV sample synthesized using Sb_2_O_3_ as catalyst did not crystallize at all on cooling. It also showed a higher *T_g_ =* 83 °C and a higher cold-crystallization temperature *T_cc_ =* 132 °C. These values are comparable to those for PET (*T_g_ =* 80 °C and *T_m_ =* 252 °C), while the *T_m_* is higher than that for PEF (*T_g_ =* 87 °C and *T_m_ =* 230 °C) [6].

The melting temperature of the cold-crystallized sample prepared with Sb_2_O_3_ was *T_m_* = 251 °C, much lower than that for the cold-crystallized sample prepared with the use of TBT (*T_m_* = 259 °C). Finally, the enthalpy of melting and crystallization was lower in the case of the higher molecular weight sample.

The cold-crystallization behavior of the polyesters was examined at different heating rates after their melt-quenching. Both samples are cold-crystallized under all the heating rates applied (Figure 4a,b). However, the lower molecular weight sample always showed a lower *T_cc_* and a larger final enthalpy, which also means a larger final degree of crystallinity. As can be seen in Figure 4c,d, the lower molecular weight sample crystallized even upon cooling at the fastest rate (20 °C/min), whilst the higher molecular weight PEV did not crystallize at all even when cooling with the slower rate of 5 °C/min. In general, the main obstacle for PEV’s crystallization proved to be poor nucleation density. This behavior is different from that of PET which always crystallizes upon cooling from the melt even at fast rates. It is rather similar to that of PEF, which also shows slow crystallization [6].

The isothermal crystallization from the melt was studied for both samples at low temperatures (large supercoolings), because, in general, the process was slow. Slow crystallization was observed for the high molecular weight sample, while under low or even moderate supercoolings both the higher and the lower *M_w_* samples could not crystallize fast (Figure 5a,b). In accordance with the previously discussed findings, the crystallization of the higher *M_w_* sample was much slower, as can be concluded given the large differences in the crystallization half-times presented in Figure 5c. In general, the crystallization half-times increased exponentially with temperature, as was expected.

To estimate the enthalpy of fusion of the pure crystalline PEV, a series of samples were prepared. First, the sample was melt-quenched. The WAXD pattern and the DSC thermogram were recorded. Afterwards, it was heated to 80 °C for 2 min, and the WAXD pattern and DSC thermogram were recorded again after cooling to room temperature. The sample was heated several times to an always higher temperature for a specific amount of time to achieve some additional crystallization. The crystallinity values were calculated from the WAXD patterns along with the relative areas under the crystalline peaks, *A_c_*, and the amorphous background, *A_am_*, using Equation (2), according to Hay et al. [45]:(2)$$X_{c}\;=\;{\left(1+\frac{A_{am}}{A_{c}}\right)}^{-1}$$

The WAXD patterns of the tested samples are presented in Figure 6a, and the heat of fusion values determined with DSC was plotted against the degree of crystallinity determined with WAXD (Figure 6b). Extrapolation to 100% degree of crystallinity resulted in a value of Δ*Η_m_* = 166 ± 16 J/g. Although the value calculated by applying the group contributions method is Δ*Η_m_* = 104 J/g [46], the value of 166 J/g seems much more reasonable, as PEV samples give in general high melting enthalpy values of about 70–100 J/g even after cold-crystallization, and this applies especially for the low *M_w_* sample.

The isothermally crystallized samples showed multiple melting behaviors upon subsequent heating, as can be seen in Figure 7a, for the low *M_w_* PEV. An exothermic recrystallization peak was observed above 200 °C. The samples crystallized at the low temperature region, below 180 °C, showed dual melting, with the recrystallization peak before final melting. The samples that crystallized above 180 °C showed triple melting. For the higher *M_w_* triple melting was observed above 160 °C (Figure 7b). Furthermore, the recrystallization exotherm is not so profound, slower than for the low *M_w_* sample and masked by the melting. In any case, it seems that the multiple melting behavior is associated with partial melting, recrystallization and final melting.

As stated above, the stage which determines PEV’s crystallization rates is nucleation, which is very slow at temperatures above 180 °C. To investigate the melting behavior after isothermal crystallization at high temperatures, crystallizations were performed after self-nucleation. As can be seen in Figure 8a, in the DSC heating traces for the low *M_w_* sample after self-nucleated crystallization at temperatures above 225 °C, multiple melting peaks appear. For samples crystallized at *T_c_* < 237.5 °C triple melting peaks are observed. Crystallization at 235.5 < *T_c_* < 245 °C results in dual melting upon subsequent heating. Finally, samples crystallized at *T_c_* > 245 °C give a single meting peak. Multiple melting is observed in DSC traces of thermoplastics after isothermal crystallization [47,48] and more often for low *M_w_* polymers [49]. It has been attributed to the melting of crystals of different stability (dual morphology mechanism) and the melting, re-crystallization, re-melting process (reorganization mechanism) [50]. Since exclusive melting of the originally formed crystals is very difficult to be observed, the origin of the multiple melting peaks appearing in DSC curves of polymers is still controversially discussed. Recent works utilizing fast chip calorimetry allow a better understanding of melting [51,52].

To investigate the melting behavior of PEV in depth, the higher *M_w_* sample was also studied, and indicative heating scans are shown in Figure 8b. Since this sample crystalized very slowly despite the prolonged crystallization times, the heat of fusion was rather low or at least much smaller than the corresponding values for the low *M_w_* sample. In any case, multiple melting behavior was observed for the higher *M_w_* sample too. To overcome uncertainty, the melting behavior of the low *M_w_* PEV after crystallization at elevated temperatures and particularly the medium melting temperature peak was used for the estimation of the equilibrium melting temperature of PEV.

#### 3.2.1. Evaluation of the Equilibrium Melting Temperature

The equilibrium melting temperature ($T_{m}^{0}\;$) is an important thermodynamic parameter for polymeric materials. The Hoffman and Weeks method [53] is one of the most commonly used for the evaluation of $T_{m}^{0}\;$. In the Hoffman and Weeks method, the measured melting points of the samples crystallized at different temperatures are plotted against the crystallization temperature and the extrapolation of the linear fit to the *T_c_ = T_m_* line, the $T_{m}^{0}$ can be estimated as the intersection point. The model is described by the Equation (3):(3)$$T_{m}\;=\;T_{m}^{0}\left(1-\frac{1}{\beta }\right)+\frac{T_{C}}{\beta }$$
where *T_m_* is the observed melting temperature of a crystal formed at a temperature *T_c_*, while *β* is the thickening parameter equal to *L_c_/L_c_^*^.* The thickening parameter *β* indicates the ratio of the thickness of the mature crystallites *L*_c_ to that of the initial ones *L_c_^*^* [53].

The Hoffman and Weeks plot for PEV was constructed for the PEV TBT, and it is presented in Figure 9. The calculated value after the extrapolation was $T_{m}^{0}$ = 301.4 °C. We used the group contributions to calculate the enthalpy and entropy of fusion [46]. Given that in equilibrium $T_{m}^{0}\;=\;\frac{\Delta {Η}_{m}}{\Delta S_{m}}$ and using the calculated values Δ*H_m_* = 20 KJ/mol = 103.6 J/g and Δ*S_m_* = 35 J/(mol K) = 0.1814 J/(g K) a value of $T_{m}^{0\;}$ = 298.4 °C was found, very close to the experimentally estimated one.

#### 3.2.2. Spherulitic Morphology and Spherulite Growth Rates

The spherulitic morphology of PEV was studied using PLM. As can be seen in Figure 10, coarsening was evidenced with increasing crystallization temperature. A clear difference can be observed above 235 °C. The spherulite growth rates were measured from the increase in the spherulite with time at various temperatures. Figure 10 shows the variation of the spherulite growth rate with temperature. A bell-shaped curve can be seen as the measurements were performed over a wide temperature range.

#### 3.2.3. Application of Secondary Nucleation Theory Using the Spherulitic Growth Rates

The spherulitic growth rate *G* data from isothermal crystallization can be analyzed with the Lauritzen-Hoffman secondary nucleation theory [54]. In the particular theory, *G* is a function of the isothermal crystallization temperature, given as follows:(4)$$G\;=\;G_{0}\;exp\left[-\frac{U^{*}}{R\left(T_{C}-T_{\infty }\right)}\right]exp\left[\frac{K_{g}}{T_{c}\left(\Delta T\right)f}\right]$$
where *G*_0_ is the pre-exponential factor. The first exponential term is associated with the impact of the diffusion process to the growth rate, while the second exponential term describes the nucleation process. *U^*^* is the activation energy for molecular diffusion across the interfacial boundary between melt and crystals, and *T*_∞_ is the temperature below which diffusion stops. *K_g_* is a nucleation parameter, while Δ*T* is the degree of undercooling (Δ*Τ* = $T_{m}^{0}$ − *T*_c_) and *f* is a correction factor which is close to unity at high temperatures and is given as *f = 2T_c_/($T_{m}^{0}$ + T_c_*).[54] The equilibrium melting point of PEV was set equal to 301.4 °C, while the glass transition was set equal to 75 °C, while the values *U^*^* = 6285 J/mol and *T**_∞_ =* (*T_g_* − 30) K were also used [54].

Prior to calculating the nucleation parameter *K_g_*, the double logarithmic transformation of Equation (4) is taken:(5)$$\ln\left(G\right)+\frac{U^{*}}{R\left(T_{C}-T_{\infty }\right)}\;=\;\ln\left(G_{0}\right)-\frac{K_{g}}{T_{c}\left(\Delta T\right)f}$$

The plot of the left-hand side of Equation (5) versus 1/*T_c_*(Δ*T*) can be fitted with a straight line, and the slope and intercept of this line give the nucleation constant and the pre-exponential factor, respectively. The critical breakpoints in the graph, which can be identified by the change in the slope of the line, indicate regime transitions accompanied by morphological changes of the formed crystals. The calculated values of the spherulitic growth rate versus temperature were used, and the Lauritzen-Hoffman plot was constructed for PEV (Figure 11). As it can be seen in the graph, a breakpoint appears at 238 °C, corresponding to regime I to regime II transition. A second breakpoint appears at about 165 °C. This corresponds to the regime II to regime III transition. These regime transitions were verified by the transitions in the spherulitic morphologies observed with PLM, as discussed above. The calculated nucleation parameter values were *K_gI_ =* 2.45 × 10^5^ K^2^, *K_gII_* = 1.19 × 10^5^ K^2^ and *K_gIII_ =* 2.74 × 10^5^ K^2^. Moreover, *K_gI_*/*K_gII_* = 2.45 × 10^5^/1.19 × 10^5^ = 2.06 that is very close to the expected *K_gI_*/*K_gII_* ratio value 2, while *K_gIII_*/*K_gII_* = 2.74 × 10^5^/1.19 × 10^5^ = 2.30. For comparison, the *K_gII_* value was found 2.50 × 10^5^ by Hay [45]. For poly(propylene terephthalate) (PPT) a *K_gII_ =* 1.47 × 10^5^ K^2^ [55] and for PBT the values *K_gII_* = 0.53 × 10^5^ K^2^ and *K_gIII_ =* 1.86 × 10^5^ K^2^ have been reported [56].

### 3.3. Thermal Degradation Kinetics

The kinetics of the thermal degradation of the PEV polymer with higher crystallinity (synthesized with TBT catalyst) were studied with TGA. Initially, a heating rate of 10 °C/min in nitrogen atmosphere was applied, in order to make a preliminary evaluation of the thermal degradation process, before fully monitoring it by means of kinetic analysis. As it can be seen in Figure 12a, the thermal decomposition of PEV is a one-step procedure, while its maximum decomposition rate is achieved at ~420 °C, corresponding to the peak of the derivative of the TGA curve. Furthermore, the rest key parameters for the evaluation of TGA measurements, which are the temperatures corresponding to 1%, 5% and 10% mass loss, are recorded at 327 °C, 373 °C and 388 °C, respectively. Finally, according to Figure 12a, PEV is not fully degraded up to 550 °C, exhibiting a residual mass of ~11% at that specific temperature.

Kinetic analysis of the thermal degradation process was carried out, in order to further investigate the thermal decomposition of PEV. According to ICTAC [57], measurements under different heating rates must be conducted to configure the rates of the procedure as a function of different variables and to investigate the degradation mechanisms thoroughly. Therefore, four heating rates were employed in the present study, namely, 5, 10, 15 and 20 °C/min (Figure 12a).

#### 3.3.1. Isoconversional Kinetics

Isoconversional (or model-free) methods of thermally induced processes provide reliable estimations concerning the activation energy (*E_α_*) values and furthermore they yield useful indications regarding the model fitting part of the kinetic study. According to the isoconversional methods, the reaction rate during a thermally stimulated process at the constant extent of conversion (*α*) is only dependent on temperature [58], and no assumption of the reaction models is required. The term extent of conversion (*α*) which is mentioned above, refers to the mass alteration when thermogravimetric measurements are employed, and it is determined as the ratio of the ongoing mass change (Δ*m*) to the entire mass loss (Δ*m_tot_*) which occurs during the total process:(6)$$\alpha \;=\;\frac{m_{0}\;-\;m}{m_{0}\;-\;m_{f}}\;=\;\frac{\Delta m}{\Delta m_{tot}}$$

In the current study, the differential method proposed by Friedman [59] and the integral method proposed by Ozawa, Flynn, and Wall (OFW) [60,61,62] are applied, in order to determine the activation energy values as a function of *α* (0.05 < *α* < 0.95).

The corresponding equation describing Friedman’s theory is the following:(7)$$ln\left[\beta_{i}{\left(\frac{d\alpha }{dT}\right)}_{\alpha ,i}\right]\;=\;ln\left[f\left(\alpha \right)A_{\alpha }\right]\;-\;\frac{E_{\alpha }}{RT_{\alpha ,i}}$$
where *I* is ascribed to the individual heating rates and *T_α,I_* to the temperature at which *α* is reached within that *i*th heating rate, whereas, the *E_α_* values are estimated from the slope of the plot of ln[*β_i_(dα/dT)_α,i_*] against (1/*T_α,i_*) [63].

Accordingly, the corresponding equation describing OFW’s theory is the following:(8)$$ln\left(\beta_{i}\right)\;=\;Const–-\;1.0516\left(\frac{E_{\alpha }}{RT_{\alpha ,i}}\right)$$
where the *E_α_* values are determined from the slope of the plot of the left part of Equation (8) against (1/*T_α,i_*)_._

The results of both methods are displayed in Figure 13, where the dependence of the *E_α_* values on the extent of conversion can be seen. The difference between the activation energies calculated by the two methods can be attributed to different factors, such as baseline stability and improper integration [64,65].

According to Figure 13, the *E_α_* − *α* dependency of both methods can be divided in two regions. At the first one (*α* < 0.9), the activation energy values are steadily increasing upon *α* augmentation, though the numerical differences are not notable. On the contrary, at high *α* values (*α*→0,95) there is a rapid increase in the *E_α_* values with increasing degree of conversion. The existence of two different regions implies the existence of two or more mechanisms that should be applied in order to successfully fit the experimental data in the model-fitting part of the kinetic analysis. Generally, in case of important differences observed in the *E_α_* values upon *α* increasing (more than 10% [66]), the model fitting part of the thermogravimetric kinetic analysis is more complex and more than one mechanisms may be required to be applied, in order to describe the decomposition of the materials. Thereafter, the great divergence in the activation energy values indicates the existence of more complicated processes, where different stages of the thermal decomposition are governed by different mechanisms.

#### 3.3.2. Model Fitting Kinetics

This part of the kinetic analysis of thermal degradation deals with the theoretical calculation of the kinetic triplet by using different models which correspond to various reaction mechanisms. In the case of a single-step process, this triplet is composed by the activation energy *E_α_* (in kJ/mol), the pre-exponential factor *A* (in s^−1^) and the reaction model *f*(*α*). That specific study is applied concurrently to the experimental curves which have been measured at different heating rates. Furthermore, the complementary character of the isoconversional and the model-fitting methods provides a criterion for the selection of the most suitable model, which is the achievement of the shortest deviation of *E_α_* values of both the aforementioned methods [63].

Initially, the decomposition of the sample is simulated by a single mechanism. In case the experimental data are poorly fitted, or the results of the fitting procedure are not in agreement with the results of the model free part of the kinetic analysis, combinations of more mechanisms should be applied. Sixteen reaction models were adopted for the single-step decomposition of PEV, with the results of the fitting procedures according to the *Cn* and *Fn* models –the most used models in the literature describing polymers’ degradation-being reported in Figure 14a,b and Table 1. It must be mentioned that *Cn* is an *n*-th order model with autocatalysis, represented by the equation: *f*(*α*) = (1−*α*)*^n^*(1+*K_cat_*.*X*), where *K_cat_* is ascribed to the autocatalysis rate constant and *X* to the reactants. Accordingly, *Fn* is an *n*-th order model, represented by the equation: *f*(*α*) = (1−*α*)*^n^.* Concerning the single mechanism fitting of PEV Figure 14a,b, both the fittings are rather poor, since the last part of the simulations presents great deviation between theoretical and experimental values. Furthermore, the values of the activation energy calculated by the single-mechanism model-fitting kinetics, namely, 166 and 178 kJ/mol for the *Cn* and *Fn* models, respectively (Table 1), are not representative of the corresponding values calculated at the isoconversional kinetic study on the whole extent of conversion *α*
Figure 13. Thus, although correlation coefficients are quite acceptable, the single mechanism fitting of PEV neither is in full agreement with the isoconversional part of the kinetic analysis, nor provides acceptable fitting quality.

Therefore, combinations of more theoretical models are required. Two-step mechanisms have been employed, with the assumption that the mechanisms are consecutive. The combinations which yielded the best fitting quality, as well as the higher correlation coefficient values for PEV, were the *n*-th order with autocatalysis in both mechanisms (*Cn–Cn*) (Figure 14c) and the *n*-th order in both mechanisms (*Fn–Fn*) (Figure 14d). The corresponding kinetic parameters are summarized in Table 1, with the correlation coefficients being high in both cases.

The values of the activation energy calculated by both models are in the proximity of the corresponding values calculated during the isoconversional kinetics study. Though, since the calculated log*K_at_* values in the *Cn*–*Cn* model are −15.47 and −6.99 (great negative numbers), and therefore, the parameter *K_cat_* is almost zero, according to the equations describing the *Cn* and *Fn* models, the *Cn* kinetic model coincides with the *Fn* model [63]. Thus, taking into consideration this remark, it is concluded that the model which best describes the thermal degradation of PEV is the consecutive *Fn*–*Fn* model.

### 3.4. Thermal Degradation Mechanism

Py–GC/MS is a valuable analytic method that allows the in-depth study of the degradation mechanisms of polymers and other complex matrices [67]. To gain insight into the specific degradation mechanism of PEV, the PEV polymer with higher crystallinity (TBT catalyst) was subjected to pyrolysis under He atmosphere at 330 °C, 420 °C and 500 °C. These temperatures were selected from the TGA data and roughly corresponded to the beginning, middle and end of degradation. The recorded total ion chromatographs (TICs) are presented in Figure 15. The main degradation products were identified through their mass spectra (not presented for brevity) and are summarized in Table 2.

Polyesters with β–hydrogen atoms degrade predominantly by heterolytic β–scission, with the transfer of the hydrogen atom from the β–alkoxy carbon atom to the ester carbonyl that results in the formation of carboxyl and vinyl end groups [11,68,69,70,71,72]. Homolytic pathways have also been identified, including acyl-oxygen and alkyl-oxygen bond scission leading to alkyl, hydroxyl and aldehyde-capped compounds. Homolysis occurs in a smaller extent, compared to β–scission and is believed to be promoted by higher pyrolysis temperatures (>300 °C) [70,71]. PEV, as a polyester with a β–hydrogen to the ester bond is expected to degrade in a similar manner, but it has a peculiarity; it contains an ether bond in para position to the ester group that is expected to result in a complex degradation mechanism because of the presence of two characteristic groups on its structure. Polymers based on vanillic acid are expected to share common features in their degradation mechanism with woody biomass, which involves homolysis, mainly of the C-O bond, and concerted routes [73,74,75,76,77]. Ethers and subsequently polyethers degrade through radical processes with mechanisms that include homolytic cleavage, disproportionation and abstraction of the formed radicals [78,79]. Therefore, a multiple mechanism that will include heterolysis and homolysis is expected to take place during the thermal degradation of PEV.

The evolution of degradation products during the pyrolysis of PEV is clearly temperature-dependent (Figure 12). At 330 °C, only a handful of compounds are detected, as evidenced by the simple chromatograph recorded. These compounds correspond to the degradation reactions that require less thermal energy to take place, providing insight into the main degradation mechanism. The number and the complexity of the peaks increase with the increase of pyrolysis temperature, leading to a final complex pattern that proves the evolution of a plethora of compounds, suggesting a complicated degradation pattern. Five main peaks were recorded in *Rt* = 15.86 min, 15.98 min, 17.22 min, 19.12 min and 24.74 min and were identified as the monomer 4-(2-hydroxyethoxy)-3-methoxybenzoic acid, ethyl 4-ethoxy-3-methoxybenzoate, methyl 4-(2-hydroxyethoxy)-3-methoxybenzoate, the methyl ester of the monomer 2-hydroxyethyl 4-(2-hydroxyethoxy)-3-methoxybenzoate and 3-methoxy-4-(2-((3-methoxy-4-(vinyloxy)benzoyl)oxy)ethoxy)benzoic acid. Their chemical structures are presented in Table 2. The nature of these compounds that contain hydroxyl, carboxyl, vinyl and methoxy end groups suggests a series of different, consequent scission reactions. More specifically, compounds containing the –CH_2_CH_2_OH and –COOH end groups of the polymer that were subjected to either C–O or C–C homolytic scission on the other end are detected, along with a β–scission derivative with *m*/*z* = 388 amu.

In higher temperatures, the end groups of all compounds remain the same, but the overall amount of degradation products increases as a result of more extensive chain scission. Additional degradation products were also detected, such as guaiacol and vanillic acid, both high added value compounds derived by the pyrolysis of biomass [80]. Noticeably, methyl 4-hydroxy-3-methoxybenzoate eluted in *Rt* ≈ 12 min in relatively large quantities, both at 420 °C and 500 °C, along with additional products with an end methoxy group, indicating extensive homolysis of the usually stable C–C bond.

It appears that heterolysis and homolysis of C–O and C–C bonds occurred simultaneously with the heterolytic scission of the ester linkage, in all pyrolysis temperatures. This hypothesis is further supported by the detection of both CO and CO_2_, which can be released after acyl-oxygen and alkyl-oxygen homolysis, respectively. The extensive homolysis could be attributed to the present of the o-methoxy group of the monomer that has been found to reduce the neighboring bond dissociation [74,75]. Additionally, the unusual extensive scission of the C–C bond could be attributed to the stabilization of the –CH_2_^·^ free radicals by the lone pairs of their adjacent oxygens.

The complexity of the degradation process concluded by Py/GC-MS is reflected in the involved pathways (Figure A1 in Appendix A) and also supported by the isoconversional and model fitting kinetics results.

### 3.5. Nanoindentation

The loading-unloading indentation curve of PEV presented a creep phenomenon at the peak force of 500 mN (Figure A2). The curve does not show any discontinuities or steps which proves no cracks were formed during the measurement. The mechanical performances of PEV are summarized in Table 3. The indentation depth at the peak load was 14.25 ± 0.26, the indentation hardness 177.81 ± 8.37 N/mm^2^ and the elastic modulus was calculated 1506.00 ± 96.19 N/mm^2^, which is close to that of polypropylene. The hardness measured for PEV lies in the same range as PET, since amorphous and annealed PET have hardness values of about 120 N/mm^2^ and up to 200 N/mm^2^, respectively [81,82].

## 4. Conclusions

In the present work, the preparation of poly(ethylene vanillate) using two different catalysts via a two-step polycondensation and followed by a solid-state polymerization step was reported. Intrinsic viscosities of 0.34 g/dL (TBT catalyst) and 0.38 g/dL (Sb_2_O_3_ catalyst) were measured for the final polymers. The polymers were characterized by DSC, WAXS and PLM. The melting, glass transition and cold-crystallization temperatures were determined and found comparable to PET. The higher molecular weight polymer (prepared with Sb_2_O_3_) exhibited a lower crystallinity. A slower crystallization was observed compared to PET, attributed to a poor nucleation density. *ΔH_m_* was estimated to be 166 ± 16 J/g and the equilibrium melting temperature 301.4 °C. The spherulitic morphology was observed with PLM; and the spherulite growth rates were measured at different temperatures and the Lauritzen-Hoffman plot constructed. A two-step mechanism was observed for the thermal degradation of PEV, which was best fitted by the *F_n_–F_n_* model (n-th order model). The thermal degradation mechanism was studied by Py/GC-MS: heterolytic scission of the ester linkages occurring in parallel to the heterolysis and homolysis of C–C and C–O bonds was evidenced. Finally, nanoindentation measurements evidenced an indentation depth at peak load of 14.25 ± 0.26 μm, an indentation hardness of 177.81 ± 8.37 N/mm^2^ and an elastic modulus of 1506.00 ± 96.19 N/mm^2^. All these data show that PEV is a promising polymer, with characteristics similar to PET, and it will be further investigated in our future work.

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
