# Peer review of "Synthesis, Thermal Properties and Decomposition Mechanism of Poly(Ethylene Vanillate) Polyester"

_polymers, 2019, doi:10.3390/polym11101672_

Round 1

Reviewer 1 Report

The aim of the study was synthesis by melt polycondensation of 4-hydroxyethylvanillic acid of poly(ethylene vanillate) (PEV) and its characteristics . The Melting enthalpy, melting entropy and equilibrium melting point were calculated for the first time. PEV thermal stability and decomposition mechanism were also investigated. The paper is an interesting study that can contribute to the scope of the Polymers. However, since the authors claim that their poly(ethylene vanillate) is a bio-based alternative to PET, Materials and Methods lack an clear statement whether their reagents are bio-based and in what percentage and finally if their PEV is 100% bio-based or not.

Thermal properties. The type of catalyst affects the molar masses obtained, and the molar masses in turn affects the thermal properties. Therefore, it is difficult to talk about the influence of the catalyst on the thermal properties, unless it remained in the polymer. Therefore, can the higher crystallinity of the lower molar mass polymer be due to residual reagents (catalyst) that act as a nucleating agents?

"PEV is not fully degraded up to 550 °C, exhibiting a residual mass of ~11% at that specific temperature". Using what catalyst the PEV test sample was obtained? The one with greater crystallinity? Were both samples tested?

The manuscript is well written and contains only spelling and typing mistakes:

Symbols representing physical quantities (or variables) such as “Tg“ should be in italic. Please correct in throughout the text. “Differential Scanning Calorimetry (DSC), Polarized Light Microscopy (PLM), and Wide-Angle X-Ray Diffraction” The apparatus names should be in lowercase. Also “Fourier Transformed-Infrared Spectroscopy”, “Nuclear Magnetic Resonance”, “X-Ray” should be X-ray, “Thermogravimetric Analysis”. Please correct in throughout the text.

Author Response

General comment: The aim of the study was synthesis by melt polycondensation of 4-hydroxyethylvanillic acid of poly(ethylene vanillate) (PEV) and its characteristics. The Melting enthalpy, melting entropy and equilibrium melting point were calculated for the first time. PEV thermal stability and decomposition mechanism were also investigated. The paper is an interesting study that can contribute to the scope of the Polymers. However, since the authors claim that their poly(ethylene vanillate) is a bio-based alternative to PET, Materials and Methods lack an clear statement whether their reagents are bio-based and in what percentage and finally if their PEV is 100% bio-based or not.

Our response:

We thank the reviewer for his/her kind words. The polymer we synthesized is not bio-based since the monomers were purchased from commercial sources. However, 15% of the present production of vanillic acid is bio-based and 2-chloroethanol can be produced from ethylene glycol which in turn can be produced from biomass. Therefore, although the polymer we synthesized is not bio-based, fully bio-based PEV can be prepared. A comment has been added in the Materials and Methods.

Comment 1: Thermal properties. The type of catalyst affects the molar masses obtained, and the molar masses in turn affects the thermal properties. Therefore, it is difficult to talk about the influence of the catalyst on the thermal properties, unless it remained in the polymer. Therefore, can the higher crystallinity of the lower molar mass polymer be due to residual reagents (catalyst) that act as a nucleating agents?

Our response:

Indeed, in both cases the catalyst remained in the polymer. Commercial polymers are produced in this fashion and therefore we conducted our studies without removing the catalysts. The catalysts could theoretically act as nucleating agents, however, their concentration is so low that their influence on crystallinity and nucleation is negligible. In fact, the lower molar mass sample was synthesized using Ti(OBu)4 which is not a nucleating agent in contrast to antimony trioxide. As stated by the reviewer, the catalysts affect the thermal properties of the polymers through their effect on the molar weight of the obtained polymers.

Comment 2: "PEV is not fully degraded up to 550 °C, exhibiting a residual mass of ~11% at that specific temperature". Using what catalyst the PEV test sample was obtained? The one with greater crystallinity? Were both samples tested?

Our response:

Thermal degradation studies were performed only on one sample: the sample prepared with TBT, which is the polymer with the greater crystallinity.

Comment 3: The manuscript is well written and contains only spelling and typing mistakes:

Our response:

The manuscript was scanned for grammatical and spelling errors and they were corrected.

Comment 4: Symbols representing physical quantities (or variables) such as “Tg“ should be in italic. Please correct in throughout the text. “Differential Scanning Calorimetry (DSC), Polarized Light Microscopy (PLM), and Wide-Angle X-Ray Diffraction” The apparatus names should be in lowercase. Also “Fourier Transformed-Infrared Spectroscopy”, “Nuclear Magnetic Resonance”, “X-Ray” should be X-ray, “Thermogravimetric Analysis”. Please correct in throughout the text. 

Our response:

We thank the reviewer for his/her thorough remark. The names have been corrected.

Reviewer 2 Report

The work of Zomboulis, Papgeorgiou and coworkers describes a highly detailed analysis of the properties of the poly(ethylene vannilate) biobased polyester. The work is performed with care, the experimental data (e.g. crystallization, degradation) is compared and reflected to models in literature. Overall the experimental approach is sound and the work provides valuable insight for others working in the field of this class of biopolymers. However, I have a few remarks that I would like to raise:

Though the polymer is called a biobased polymer, the authors make use chemicals such as 2-chloroethanol. Are these monomers also considered biobased, if so then I think it would be appropriate to mention this in the introduction.

4-hydroxyethyl vanillic acid or 4-(2-hydroxyethoxy)-3-methoxybenzoic acid? Both are used in the text.

The PEV is compared to PET and PEF and is called a polyester. However, when looking at the chemical structure provided in Figure 1, I consider poly(ether-ester) a better name for this class of polymers. In particular, with respect to stability and mobility of the polymer chains, fundamental differences can be expected.

The main problem with the current work that has to be addressed is the definition of the high and low molecular weight polymers. In the introduction, the authors mention previous work on PEV by Mailon et al, and Gioia et al (working with materials in the range of ~5000 g/mol). Though the authors have performed IV studies, no information on the actual molecular weight is available. How does the material of the authors compare with those in literature? I consider this very important as crystallization behavior is highly dependent on the molecular weight of these relatively low molecular weight materials (as also highlighted by the authors). I would suggest the authors to perform size-exclusion chromatography experiments to better describe the molecular weight properties of their materials and ease the comparison with literature, also taking into account people who would want to compare their future work to the data provided in the current manuscript.

The manuscript provides detailed analysis but is also rather lengthy and dense in information. My suggestion would be to provide the degradation information and nano-indentation information in a supporting information section. This would, in my opinion, sharpen the focus of the work.

Author Response

General comment: The work of Zomboulis, Papgeorgiou and coworkers describes a highly detailed analysis of the properties of the poly(ethylene vannilate) biobased polyester. The work is performed with care, the experimental data (e.g. crystallization, degradation) is compared and reflected to models in literature. Overall the experimental approach is sound and the work provides valuable insight for others working in the field of this class of biopolymers. However, I have a few remarks that I would like to raise:

Our response:

We thank the reviewer for his/her kind words.

Comment 1: Though the polymer is called a biobased polymer, the authors make use chemicals such as 2-chloroethanol. Are these monomers also considered biobased, if so then I think it would be appropriate to mention this in the introduction.

Our response:

2-chloroethanol can be produced from ethylene glycol which can be produced from biomass. Therefore, a fully bio-based PEV can potentially be prepared. The introduction has been modified to underline the latent bio-based character of 2-chloroethanol.

Comment 2: 4-hydroxyethyl vanillic acid or 4-(2-hydroxyethoxy)-3-methoxybenzoic acid? Both are used in the text.

Our response:

Both names are correct. 4-(2-hydroxyethoxy)-3-methoxybenzoic acid is using the IUPAC nomenclature for organic compounds, while 4-hydoxyethyl vanillic acid is based on the trivial name of 4-hydroxy-3-methoxybenzoic acid which is vanillic acid. 4-hydroxyethyl vanillic acid has been changed to 4-(2-hydroxyethoxy)-3-methoxybenzoic acid.

Comment 3: The PEV is compared to PET and PEF and is called a polyester. However, when looking at the chemical structure provided in Figure 1, I consider poly(ether-ester) a better name for this class of polymers. In particular, with respect to stability and mobility of the polymer chains, fundamental differences can be expected.

Our response:

Th reviewer is absolutely right, however we chose the term “polyester” to be consistent with the previously published literature (for example the works of Mialon et al. [1] or Celli et al. [2]). Additionally, the term poly(ether-ester) is often used for block copolymers or polymers containing polyether segments which is not our case.

Mialon, L.; Vanderhenst, R.; Pemba, A.G.; Miller, S.A. Polyalkylenehydroxybenzoates (PAHBs): Biorenewable aromatic/aliphatic polyesters from lignin. Macromol. Rapid Commun. 2011, 32, 1386–1392. Gioia, C.; Banella, M.B.; Marchese, P.; Vannini, M.; Colonna, M.; Celli, A. Advances in the synthesis of bio-based aromatic polyesters: Novel copolymers derived from vanillic acid and ϵ-caprolactone. Polym. Chem. 2016, 7, 5396–5406.

Comment 4: The main problem with the current work that has to be addressed is the definition of the high and low molecular weight polymers. In the introduction, the authors mention previous work on PEV by Mailon et al, and Gioia et al (working with materials in the range of ~5000 g/mol). Though the authors have performed IV studies, no information on the actual molecular weight is available. How does the material of the authors compare with those in literature? I consider this very important as crystallization behavior is highly dependent on the molecular weight of these relatively low molecular weight materials (as also highlighted by the authors). I would suggest the authors to perform size-exclusion chromatography experiments to better describe the molecular weight properties of their materials and ease the comparison with literature, also taking into account people who would want to compare their future work to the data provided in the current manuscript.

Our response:

We thank the reviewer for this very pertinent remark. Unfortunately, size-exclusion chromatography experiments require specialized solvents such as hexafluoropropanol and we do not have the possibility to carry out such experiments.

Comment 5: The manuscript provides detailed analysis but is also rather lengthy and dense in information. My suggestion would be to provide the degradation information and nano-indentation information in a supporting information section. This would, in my opinion, sharpen the focus of the work. 

Our response:

Figures 16 and 17 have been moved to Appendix A.

Reviewer 3 Report

The manuscript submitted by Bikiaris and Papageorgiou reports on poly(ethylene vanillate) (PEV), a recently known bio-based polyester with high potential in the replacement of commercial semiaromatic polyesters of petrochemical origin. As it happens with other novel bio-based polymers, there are important gaps in the structure-properties knowledge of PEV that need to be filled. Although some synthesis is reported in this paper, the most relevant contribution is the valuable information it affords regarding the thermal behavior of PVE including melting-crystallization transitions and heat decomposition. A good amount of DSC and TGA data with their corresponding sound interpretation is provided in full detail using well-applied methodologies. It is very likely that the information contained in this paper will be a frequent reference for following work that will appear on PEV in the near future. The manuscript is well organized and written and in my opinion it could be published essentially as it is. Nevertheless, a few suggestions are made for the authors wish to consider them for some improvement. a) In the Abstract, the paragraph “The enthalpy of fusion, the entropy of fusion and the equilibrium melting temperature have been calculated for the first time. Additionally, the isothermal and non-isothermal crystallization behavior are described and new insights on the thermal stability and degradation mechanism of PEV are given” could be rewritten to give major importance to the crystallization and thermal stability studies. As it is written, it seems that temperatures measurements, which is frequently considered a routinely task, are the main issue. b) The paper is a bit long. I would suggest removing some non-essential figures (e.g. some spherulites optical microscopy pictures) and shortening some text details regarding crystallization kinetics methodology and discussion of the heat decomposition results. c) Some grammatical/typo errors have been detected so that a slight revision of the language could be convenient.

Author Response

General Comment: The manuscript submitted by Bikiaris and Papageorgiou reports on poly(ethylene vanillate) (PEV), a recently known bio-based polyester with high potential in the replacement of commercial semiaromatic polyesters of petrochemical origin. As it happens with other novel bio-based polymers, there are important gaps in the structure-properties knowledge of PEV that need to be filled. Although some synthesis is reported in this paper, the most relevant contribution is the valuable information it affords regarding the thermal behavior of PVE including melting-crystallization transitions and heat decomposition. A good amount of DSC and TGA data with their corresponding sound interpretation is provided in full detail using well-applied methodologies. It is very likely that the information contained in this paper will be a frequent reference for following work that will appear on PEV in the near future. The manuscript is well organized and written and in my opinion it could be published essentially as it is. Nevertheless, a few suggestions are made for the authors wish to consider them for some improvement.

Our response:

We appreciate the very kind words of the reviewer and thank him/her for his/her suggestions.

Comment a): In the Abstract, the paragraph “The enthalpy of fusion, the entropy of fusion and the equilibrium melting temperature have been calculated for the first time. Additionally, the isothermal and non-isothermal crystallization behavior are described and new insights on the thermal stability and degradation mechanism of PEV are given” could be rewritten to give major importance to the crystallization and thermal stability studies. As it is written, it seems that temperatures measurements, which is frequently considered a routinely task, are the main issue.

Our response:

The paragraph has been rephrased taking into account the comment of the reviewer and pointing out that the thermal quantities in question refer to pure crystalline PEV and thus are not just routine temperature measurements.

Comment b): The paper is a bit long. I would suggest removing some non-essential figures (e.g. some spherulites optical microscopy pictures) and shortening some text details regarding crystallization kinetics methodology and discussion of the heat decomposition results.

Our response:

The size of the microscopy figures was decreased. Additionally, figures 16 and 17 have been moved to Appendix A.

Comment c): Some grammatical/typo errors have been detected so that a slight revision of the language could be convenient.

Our response:

Grammatical and spelling errors were corrected.

Round 2

Reviewer 1 Report

Thank you for considering most of the comments. However, if the polymer is not bio-based, the authors should remove the word bio-based from the title. For readers, this is misleading.

Please provide in the text information that the thermal degradation studies were carried out on only the sample prepared with TBT, which is the polymer with the greater crystallinity.

Symbols representing physical quantities (or variables) such as “Tg“ and “Tm” should be in italic. Please correct in throughout the text.

Author Response

General comment: Thank you for considering most of the comments. However, if the polymer is not bio-based, the authors should remove the word bio-based from the title. For readers, this is misleading.

The word “bio-based” has been removed from the title.

Comment 1: Please provide in the text information that the thermal degradation studies were carried out on only the sample prepared with TBT, which is the polymer with the greater crystallinity.

The text has been modified:

Line 432: The kinetics of the thermal degradation of the PEV polymer with higher crystallinity (synthesized with TBT catalyst) were studied with TGA.

Line 534: To gain insight into the specific degradation mechanism of PEV, the PEV polymer with higher crystallinity (TBT catalyst) was subjected to pyrolysis under He atmosphere at 330 °C, 420 °C and 500 °C.

Comment 2: Symbols representing physical quantities (or variables) such as “Tg“ and “Tm” should be in italic. Please correct in throughout the text.

Symbols have been corrected.

Reviewer 2 Report

The authors have addressed all points raised, albeit without performing additional experimental work. However, given that the authors have no access to equipment required to perform the suggested experiments, the report experimental approach is deemed sufficient for other to reproduce the work. 

Author Response

We thank the reviewer for his/her kind words and for accepting the paper for publication.